# Effects of Self-Efficacy and Outcome Expectations on Motor Imagery-Induced Thermal and Mechanical Hypoalgesia: A Single-Blind Randomised Controlled Trial

**DOI:** 10.3390/ijerph191911878

**Published:** 2022-09-20

**Authors:** Ferran Cuenca-Martínez, Elena Bocos-Corredor, África Espinosa-Giménez, Laura Barrero-Santiago, Naira Nefa-Díaz, David Canchal-Crespo, Clovis Varangot-Reille, Aida Herranz-Gómez, Luis Suso-Martí, Núria Sempere-Rubio, Roy La Touche

**Affiliations:** 1Faculty of Psychology and Education Sciences, Universitat Oberta de Catalunya, 08018 Barcelona, Spain; 2Exercise Intervention for Health Research Group (EXINH-RG), Department of Physiotherapy, University of Valencia, 46010 Valencia, Spain; 3Departamento de Fisioterapia, Centro Superior de Estudios Universitarios La Salle, Universidad Autónoma de Madrid, 28049 Madrid, Spain; 4UBIC, Department of Physiotherapy, Faculty of Physiotherapy, Universitat de València, 46010 Valencia, Spain; 5Motion in Brains Research Group, Institute of Neuroscience and Sciences of the Movement (INCIMOV), Centro Superior de Estudios Universitarios La Salle, Universidad Autónoma de Madrid, 28049 Madrid, Spain; 6Instituto de Neurociencia y Dolor Craneofacial (INDCRAN), 28003 Madrid, Spain

**Keywords:** pain expectations, motor imagery, self-efficacy, outcome expectations, pain modulation

## Abstract

The main aim of this study was to assess whether self-efficacy (SE) and outcome expectations (OEs) modulate the hypoalgesic effect induced by motor imagery (MI). A total of 75 asymptomatic participants were randomly assigned to the positive (SE+, OE+), negative (SE−, OE−) or non-expectation (CG) groups. Heat pain threshold (HPT) and pain pressure threshold (PPT) were the main variables. Cold detection threshold (CDT), warm detection threshold (WDT), heart rate (HR) and perceived fatigue were the secondary variables. The variables were assessed preintervention, immediately postintervention and 10 min postintervention, except for HR, which was measured continuously during the intervention. Regarding HPT, significant within-group pre-post differences were found in the OE+ group, with a low effect size (*p* = 0.01, *d* = −0.39). With regard to ΔPPT, significant intergroup differences were found in Δpost-pre between the SE+ and CG groups (*p* = 0.012, *d* = 1.04) and also between SE+ and OE− (*p* = 0.006, *d* = 1.08), both with a large effect size. CG, SE−, and OE− groups had poorer CDT and WDT. Regarding HR, significant intergroup differences were found in the postintervention measurement between OE+ and SE−, with a large effect size (*p =* 0.016, *d* = 1.34). Lastly, no between-group differences were found regarding perceived fatigue (*p* > 0.05). The results obtained showed that positive expectations have a slight influence on the increase in heat and mechanical pain detection thresholds. Positive and non-expectancy groups showed an autonomic activation. The results also showed that negative expectations led to poorer perceptual processes.

## 1. Introduction

Movement representation strategies appear to elicit hypoalgesic effects both in isolation [1] and in combination with physical movement [2]. There is a level of evidence of 1 that this hypoalgesia occurs when they are combined with them or with other interventions [3]. Beinert et al. [4] found no difference in hypoalgesia generated by imagining and physically performing cervical motor control exercises. The study by Suso-Martí et al. [1] indicates that distraction is likely to be a mechanism underlying the hypoalgesic effects mediated by movement representation strategies. Distraction was also noted by Hayashi et al. [5] and Peerdeman et al. [6]. However, it appears that there might be several additional mechanisms that can explain the hypoalgesic effects of motion representation strategies. For example, Larsen et al. [7] showed that motor imagery (MI) and action observation training could increase in cortical excitability, which was associated with a reduction in pain perception. Additionally, it is likely that the autonomic nervous system partly explains the hypoalgesia generated by movement representation strategies [1].

Bialosky et al. [8] observed that expectations appeared to modulate the hypoalgesic effect of manual therapy techniques. More recently, Vaegter and Jones [9] commented that expectations should be assessed before prescribing exercise in patients with persistent pain because of the influence that expectations can have on exercise-induced hypoalgesia. Expectations were defined by Bandura [10], who outlined the difference between self-efficacy (SE) and outcome expectations (OE). SE is the personal conviction that one can successfully execute the behaviour required to produce an outcome; and OE is defined as a person’s estimate that a given behaviour will lead to certain outcomes [10]. Expectations influence conscious physiological processes, such as pain and motor performance [11]. Assuming that movement representation strategies can elicit hypoalgesic effects and also that expectations can modulate the hypoalgesic effect of clinical interventions, such as manual therapy or therapeutic exercise, we sought to determine whether expectations would modulate the hypoalgesic effect elicited by movement representation strategies. To date, no study has verified these effects. Therefore, the main aim of this study was to assess whether self-efficacy and outcome expectations are able to modulate the hypoalgesic effect induced by motor imagery.

## 2. Materials and Methods

### 2.1. Study Design

We conducted a single-blind randomised controlled trial whose protocol followed the Consolidated Standards of Reporting Trials statement on randomised trials of nonpharmacological treatments [12].

### 2.2. Participants

A sample of 75 asymptomatic volunteers was recruited from the local community through social media and e-mail. Participants were recruited between 1 December 2020 and 31 May 2021. The inclusion criteria were as follows: asymptomatic participants (pain free), aged 18 to 65 years, and have an absolute understanding of the language. The exclusion criteria were the following: any knowledge of physical therapy, psychology, or movement representation techniques; age younger than 18 years; pain at the time of the study; or any type of neurological or musculoskeletal disease. All procedures were approved by the Human Research Ethics Committee of the La Salle University Centre for Advanced Studies (CSEULS-PI-048/2020). The study was registered in the United States Randomised Trials Register on clinicaltrial.gov (NCT04621162). All participants granted their written informed consent prior to inclusion and were provided an explanation of the study procedures, which were planned under the ethical standards of the Helsinki Declaration.

### 2.3. Randomisation

Randomisation was performed using a computer-generated random sequence table with a balanced 5-block design (GraphPad Software, Inc., San Diego, CA, USA). An independent researcher generated the randomisation list, and a research team member who was not involved in the assessment or intervention of the participants performed the randomisation and maintained the list. Those included were randomly assigned to one of the 5 study groups using the random sequence list, ensuring concealed allocation.

### 2.4. Blinding

The assessments and interventions were performed by different researchers. The evaluator was blinded to the participant’s assignment when performing the measurements and recording the data. The participants were asked not to make any comments to the researcher performing the measurements. Therefore, the evaluator was unaware of the intervention that each participant had received.

### 2.5. Intervention

First, the resting heart rate was measured just before the intervention. During the intervention, the heart rate was measured every 15 s, similar to Cuenca-Martínez et al. or Decety et al. [13,14]. At rest, no heart rate data were collected. The participants were randomly assigned to receive a positive or a negative self-efficacy expectation (SE+, SE−, OE+ and OE−). The control group (CG) did not receive any expectation. The researcher in charge of giving the expectations had a 3 h training with a psychologist from the University to give them in the same way every time (intonation, rhythm, voice volume, gestures, etc.) in order to normalise and control the intervention. Participants in the positive expectation group were told the following:

SE+: *‘According to the results of the initial evaluation, you will be very capable of performing the imagination task.’*

OE+: *‘According to the evidence, this technique is a very effective way to treat pain in physical therapy and we expect it will reduce your perception of pain in evaluation tests.’*

Participants in the negative expectation group were told the following:

SE−: *‘According to the results of the initial evaluation, you are not going to be able to perform the imagination task.’*

OE−: *‘According to the evidence, this technique is an ineffective way to treat pain in physical therapy and we expect it will increase your perception of pain in evaluation tests.’*

The participants had to imagine themselves performing 3 kinaesthetic movements from a first-person perspective (Figure 1). These tasks had to be accompanied by feeling the movement during the imagination process. Participants had their eyes closed during the entire intervention. Major emphasis was placed on sensory references to try to ensure maximum kinaesthetic MI. The vertical jump was the first movement to imagine. They had to imagine themselves from the standing position performing a high jump toward the ceiling, raising their arms. The next movement was running. They were told to imagine themselves running as they usually run. In the third exercise, they were told to imagine performing a squat lowering themselves to the ground taking a rod.

The intervention lasted 8.5 min. Other research studies with movement representation techniques had similar durations because continuous imagination can lead to mental fatigue [15,16]. For each exercise, the participants imagined for 1 min, rested for 30 s, and returned to imagining the same task for another minute. They were told that no real movement should be made during the intervention. The participant’s task was accompanied by the following phrases: ‘keep imagining’, ‘focus your attention on the thigh’, ‘remember to imagine the task in the first person’, and ‘try to feel the movement’. In addition, specific instructions were given according to the task to be imagined. The prompts were identical for all participants regardless of the group to which they belonged. At the end of the intervention, each participant rated their mental fatigue on a visual analogue scale (VAS-f).

### 2.6. Outcome Measures

#### 2.6.1. Primary Outcomes

##### Heat Pain Threshold (HPT)

The thermal quantitative sensory test was performed using a computerised thermal stimulator (MEDOC TSA-2001 apparatus, Medoc Ltd., Ramat-Yishai, Israel) [17,18]. The thermo sensor, which has a size of 30 × 30 mm, was placed on the vastus medialis (L3 dermatome). The contact area of the thermode was 9.0 cm^2^. Participants were instructed to press a button on a remote-control device as soon as they perceived heat pain, following the instructions given by the evaluator. The temperature of the thermode started at a baseline of 32 °C and heated up at a rate of 1 °C/s to the upper limit of 50 °C. The mean thresholds of 3 consecutive measurements were calculated. The procedure then ended, and the temperature returned to baseline. The participants were instructed not to look at the computer screen at any time during the testing procedures. The entire protocol was based on the study by Wang et al. [17]. HPT (ICC = 0.64–0.88) showed good to excellent reliability.

##### Pressure Pain Threshold (PPT)

The PPT is defined as the minimal amount of pressure at which a sense of pressure first changes to pain [19]. The mechanical pressure algometer (Wagner Instruments, Greenwich, CT, USA) used in this study consisted of a round rubber disk (area 1 cm^2^) attached to a pressure (force) gauge. The gauge displays values in kilograms, but because the surface of the rubber tip is 1 cm^2^, the readings are expressed in kg/cm^2^. The pressure algometer values range from 0 to 10 kg, in 0.1 kg intervals. Pressure was applied at a rate of 0.31 kg/s [20]. The reliability of the pressure algometry was high (ICC = 0.91 [95% confidence intervals (CI): 0.82–0.97] [20]. PPTs were tested in 2 locations: the quadriceps femoris muscle and the upper trapezius muscle. All the assessments were performed in a quiet room. Three consecutive measurements were taken of the PPT at the 2 locations at intervals of 30 s, and the mean of these three trials was used for the data analysis [21].

#### 2.6.2. Secondary Outcome Measures

##### Cold and Warm Detection Threshold

Thermal quantitative sensory tests were also performed using a computerised thermal stimulator (MEDOC TSA-2001 apparatus, Medoc Ltd., Ramat-Yishai, Israel) [17,18]. The participants were instructed to press a button on a remote-control device as soon as they perceived the thermal sensation of cold and warm, following the instructions given by the evaluator. The temperature of the thermode started at a baseline of 32 °C and cooled down or heated up at a rate of 1 °C/s to a lower limit of 20 °C or upper limit of 50 °C. The mean thresholds of 3 consecutive measurements were calculated. The 3 thermal quantitative sensory tests were measured at once, with an interval of 4 to 10 s, which was the time required to return to baseline. The procedure then ended, and the temperature returned to the baseline. The participants were instructed not to look at the computer screen at any time during the testing procedures. The entire protocol was also based on the study by Wang et al. [17]. The cold-detection threshold (CDT) (ICC = 0.78–0.94) showed good to excellent reliability, and warm detection threshold (WDT) was fair to excellent (ICC = 0.23–0.69).

##### Autonomic Nervous System Measurement

Heart rate (HR) was measured to evaluate the ANS response as Suso-Martí et al. [1] carried out. The HR was recorded to quantify the changes produced in the ANS during the performance of the mental practice. The Garmin Forerunner VR 225 is a commercially available wrist-worn HR monitor that uses an optical green light sensor to detect HR. The Garmin Forerunner VR 225 was programmed with the participants’ sex, age, weight and height and was fitted on the left forearm according to the user manual. Previous studies have shown moderate to strong validity of the Garmin Forerunner VR 225 versus traditional ECG measures (Pearson *r* = 0.650–0.868).

##### VAS on Fatigue

We employed the VAS-f to quantify the participants’ perceived fatigue after performing the training session. The VAS-f uses a line of 100 mm, with 0 representing minimum fatigue (no fatigue) and 100 representing maximum fatigue. The VAS-f scale is useful, sensitive, and easy to apply [22].

#### 2.6.3. Baseline Outcomes

##### Mental Imagery Ability

To assess motor imagery ability, we employed the Movement Imagery Questionnaire-Revised (MIQ-R), which consists of 4 movements repeated in 2 domains (visual and kinaesthetic). Depending on the perceived difficulty, participants score the movements from 1 to 7, with 1 representing the maximum difficulty in creating mental motor imagery and 7 representing the least difficulty. We extracted the total score and the score of the visual and kinaesthetic subscale. The psychometric properties of the MIQ-R were consistently adequate, with Cronbach’s α coefficients ranging above 0.84 for the entire scale, 0.80 for the visual domain and 0.84 for the kinaesthetic domain [23].

##### Mental Chronometry

Mental chronometry (MC) is a reliable and widely used tool for objective measurements of the ability to create mental motor images [24]. For the MC assessment, we used a stopwatch to record the time spent by each participant on imagining the mental tasks in the MIQ-R. The evaluator issued a command to start imagining the task, and the participant performed a verbal sign once the task had been completed. The time between the two interval commands was recorded, as was the time dedicated by each participant to the real-time execution of the task. The MC values are expressed as the time congruence between the 2 tasks. The inter-rater intraclass correlation coefficient (ICC) for MC ranged from 0.63 to 0.95, whereas the ICC for intrasession reliability ranged from 0.95 to 0.97 [24].

##### Physical Activity Level

We employed the self-reported International Physical Activity Questionnaire (IPAQ) to assess the participants’ physical activity level. The total estimated resting expenditure energy (W/m^2^), or MET, was extracted. The questionnaire’s psychometric properties were accepted for use in studies that measure physical activity; the IPAQ has a reliability of approximately 0.65 (r = 0.76; 95% CI 0.73–0.77) [25].

### 2.7. Procedures

After giving their consent to partake in the study and prior to the intervention, all participants were given a set of questionnaires, including a sociodemographic assessment and an evaluation of their physical activity, MC and ability to imagine movements. The evaluator gave the same explanation of the procedure to all participants. Firstly, a pre-measurement of the thermal quantitative sensory tests and the PPT was performed. Then, the intervention was explained to the participants, and they were informed that their HR would be monitored by an HR monitor during the intervention. At this point, the intervention was carried out, which is amply detailed in the section ‘intervention’ (*see above*). After the intervention, the same variables were measured immediately after and at 10 min post-intervention.

### 2.8. Sample Size Calculation

The sample size was estimated with the program G*Power 3.1.7 for Windows (G*Power© from University of Dusseldorf, Germany) [26]. The sample size calculation was considered as a power calculation to detect between-group differences in a primary outcome measure (HPT). We considered 5 groups (SE+, OE+, SE−, OE− and CG) and 3 measurements (pre-, post- and 10 min post-intervention) for primary outcomes to obtain 95% statistical power (1-β error probability) with an α error level probability of 0.05 using analysis of variance (ANOVA) of repeated measures, within-between interaction, and an effect size of η*p*^2^ = 0.175 obtained from our results. This generated a sample size of total of 65 participants plus an estimated 20% loss in follow-up, yielding a total of 75 participants (15 per group).

## 3. Results

A total of 75 asymptomatic participants were included in the study and were randomly assigned to 5 balanced groups consisting of 15 participants per group. There were no adverse events or dropouts reported in any of the groups. All variables showed normal distribution. No statistically significant differences were found in the baseline measurements between the five groups (*p* > 0.05), except for the quadriceps muscle PPT (*p* < 0.01) (Table 1). For this reason, increments were calculated for this outcome measure (PPT).

### 3.1. Primary Outcomes

#### 3.1.1. Heat Pain Threshold

The ANOVA revealed significant changes in HPT over time (*F =* 6.25, *p* = 0.006, η^2^ = 0.082) but not during the time*group interaction (*F =* 0.94, *p* = 0.465, η^2^ = 0.05). The post hoc analysis revealed significant within-group, pre-post differences in the OE+ group with a low effect size (*p* = 0.01, *d* = −0.39). No differences were found in the OE−, CG, SE− and SE+ groups. Finally, the post hoc analysis revealed no significant between-group differences (*p* > 0.05) (Figure 2).

#### 3.1.2. Change in Pain Pressure Threshold

The ANOVA revealed significant changes in ΔPPT over time (*F* = 12.13, *p =* 0.001, η^2^ = 0.14) and also during time*group interaction (*F* = 7.41, *p* < 0.001, η^2^ = 0.29). The post hoc analysis revealed significant inter-group differences in Δpost-pre between SE+ and CG (*p =* 0.012, *d* = 1.04) and also between SE+ and OE− (*p =* 0.006, *d* = 1.08), both with a large effect size. However, the post hoc analysis did not reveal significant between-group in Δpost-10 min-pre PPT differences (*p* > 0.05). Lastly, the post hoc analysis revealed significant within-group Δpost-pre and Δpost-10 min pre differences in the SE+ group with a moderate effect size (*p <* 0.001, *d* = 0.74) (Figure 3).

#### 3.1.3. Distal Pain Pressure Threshold

The ANOVA revealed no significant changes in the trapezius PPT over time (*F* = 0.02, *p* = 0.945, η^2^ = 0.00) nor in the during time*group interaction (*F =* 0.45, *p* = 0.80, η^2^ =0.02).

### 3.2. Secondary Outcomes

#### 3.2.1. Cold Detection Threshold

The ANOVA analysis revealed significant changes in CDT over time (*F* = 20.02, *p* < 0.001, η^2^ = 0.222) but not during the time*group interaction (*F* = 2.11, *p* = 0.088, η^2^ = 0.108). The post hoc analysis revealed significant within-group pre-post and pre-post-10 min differences in the CG, with a large effect size (*p* < 0.001, *d* = 0.83 and *p* = 0.004, *d* = 0.85, respectively), and also in both negative expectation groups (OE− and SE−), with a moderate effect size (*p* < 0.001, *d* = 0.68 and *p =* 0.016, *d* = 0.72 for the OE− group and *p* = 0.007, *d* = 0.56 and *p* = 0.05, *d* = 0.51 for the SE− group). No differences were found in the OE+ and SE+ groups (Table 2). Finally, the post hoc analysis did not reveal significant between-group differences (*p* > 0.05).

#### 3.2.2. Warm Detection Threshold

The ANOVA analysis revealed significant changes in the WDT over time (*F* = 26.03, *p <* 0.001, η^2^ = 0.271) but not in the time*group interaction (*F =* 1.84, *p* = 0.09, η^2^ = 0.095). The post hoc analysis revealed significant within-group, pre–post-10 min differences in the CG, with a large effect size (*p* = 0.01, *d* = −1.04), and also in both negative expectation groups, with a large effect size (*p* < 0.001, *d* = −0.98 for the OE− group and *p* = 0.001, *d* = −0.81 for the SE− group). In addition, the post hoc analysis revealed significant within-group post–post-10 min (immediately after the intervention and 10 min later) differences in the OE− group, with a moderate effect size (*p* < 0.001, *d* = −0.51) and significant within-group pre-post differences in the SE−, with a moderate effect size (*p <* 0.015, *d* = −0.76). No differences were found in the OE+ and SE+ groups (Table 2). Finally, the post hoc analysis did not reveal significant between-group differences (*p* > 0.05).

#### 3.2.3. Heart Rate

The ANOVA revealed significant changes in HR over time (*F* = 15.32, *p* < 0.001, η^2^ = 0.18) and also in the time*group interaction (*F* = 3.37, *p* = 0.003, η^2^ = 0.162). The post hoc analysis revealed significant inter-group differences in post-intervention measurements between the OE+ and SE− groups, with a large effect size (*p =* 0.016, *d* = 1.34). In addition, the post hoc analysis revealed significant within-group differences in OE+ group between pre-mid and pre-post intervention (*p* = 0.001, *d* = −0.56 and *p* =< 0.001, *d* = −1.01, respectively) and also in the SE+ and CG groups between pre-post intervention measurements, with a moderate effect size (*p* = 0.006, *d =* −0.50, and *p* < 0.001, *d* = −0.55, respectively) (Figure 4).

#### 3.2.4. Fatigue Visual Analogue Scale

The one-way ANOVA did not reveal significant between-group differences regarding perceived fatigue (*F =* 2.45, *p =* 0.054).

## 4. Discussion

The main aim of this study was to assess whether the SE and OE groups were able to modulate the hypoalgesia effect induced by MI. The results obtained showed, first, that positive expectations have a slight influence on the improvement of heat and mechanical pain detection thresholds, where even in the latter, significant between-group differences between positive and negative expectations were observed. It appears that HPTs are more robust than PPTs and therefore, they are less susceptible to modulation by expectations. These findings were accompanied by neurophysiological correlates, in that the two positive expectations groups showed a statistically significant increase in HR, and this sympathetic activation did not occur in the negative expectancy groups (which did not produce hypoalgesic effects in these groups). However, the no-expectation group also showed significant intra-group differences.

MI involves cognitive processes such as working memory or attention [15]. These cognitive processes have high requirements that are capacity limited [27]. We might need an incentive to engage in a high level of mental effort on a single task [28]. Locke and Braver found that the presence of a reward improved the achievement of a cognitive task, through sustained increased activation of cortical areas was related to working memory and cognitive control [29]. Moreover, the presence of a motivational stimulus also influences attention-related areas and sensory cortices [30,31]. Indeed, Frömer et al. found that participants allocated more cognitive control to a task when they expected a reward and/or expected their behaviour to have efficacy [32]. In our study, we found that when we provided positive expectations (pain relief as a reward or a positive self-efficacy expectation) to the participants, they had a greater statistically significant increase in their heart rate during the MI task, a factor previously observed with mental effort [14,33,34]. This increase was not present in the neutral or negative expectation groups. Participants in the negative conditions might not have allocated sufficient mental effort to this specific task due to an expected punishment or expected poor efficacy. Patients in the neutral conditions might not have allocated sufficient mental effort during MI because they had no knowledge of its possible effectiveness. Our study is the first to demonstrate the importance of providing positive expectations prior to MI to stimulate the participant to allocate sufficient cognitive control to the MI. Future studies should evaluate the effect of positive expectations (self-efficacy and/or outcomes) on other well-studied variables, such as motor learning [35].

We found that positive or negative expectations minimally modulated the hypoalgesic effect of MI in asymptomatic patients. What is relevant here is that the intervention was exactly the same for all groups. Various authors have shown the pain-relief effect of positive expectations and the increased pain effect of negative expectations in nonpharmacological treatments [8,9]. One hypoalgesic mechanism could be a direct effect of expectations on nociceptive processing. According to the predictive coding theory, perceived pain is estimated from a prior expectation of pain and observation of the current nociceptive input [36]. Thus, when we provide positive cues to the participant, prior expectations will be that perceived pain will be less [36]. Placebo hypoalgesia takes place through the activation of the μ-opioid and dopaminergic reward-related systems, among others, in the cortical, subcortical and brainstem structures [37].

Another mechanism for the expected hypoalgesic effect could be more indirect: expectations might modulate nociceptive processing via the improvement in the achievement of MI. As noted earlier, a motivational stimulus improves the achievement of a cognitive task [32]. MI activates neural networks that are also activated during the real movement [38]; thus, they might share a hypoalgesic effect. We found that only positive expectations enhanced autonomic responses and that only these groups had an increase in their pain threshold. Future studies should further examine the neurophysiological mechanisms behind the hypoalgesic effect of positive expectations in MI.

We found that the neutral condition had no hypoalgesic effect on mechanical pain thresholds or on the heat pain threshold. La Touche et al. found a hypoalgesic effect on the mechanical pain threshold after three sessions of 45 min of motor imagery orofacial training [39]. Morales-Tejera et al. [40] also found an increase in the mechanical pain threshold after five sessions of 20 min of MI cervical training. Our intervention had a shorter duration (only 8.5 min) and included only one session, which might not have been sufficient to elicit a hypoalgesic effect on asymptomatic participants. We want to highlight that positive outcome expectations did modulate the thermic pain threshold, and positive self-efficacy expectations modulated the mechanic pain threshold. In addition to possibly providing various neurophysiological effects on nociceptive processing, adding positive expectations might be a way to perceive hypoalgesic effects with less intervention. Moreover, Frömer et al. observed a statistically significant interaction between high reward and high efficacy on task performance [32]. Futures studies should consider combining both types of positive expectation to improve the MI hypoalgesic effects. Finally, in peripheral tissues, noxious mechanical, thermal, and chemical stimuli are detected by nociceptors and converted to action potentials by lightly myelinated Aδ and nonmyelinated C-fibers. Unlike other sensory receptors, most nociceptors are thought to be polymodal (i.e., sensitive to both heat and mechanical noxious stimuli). Until recently, discrimination between different pain modalities was therefore considered to happen primarily at the spinal and/or supraspinal sites [41]. Indeed, recent findings revealed that noxious stimuli of various natures specifically activate different neuronal pathways and, consequently, that the distinction between pain modalities would instead occur at the level of primary afferents. In particular, this model suggests that heat and mechanical sensitivity are processed by distinct subpopulations of primary afferent fibres [42].

Thermal pain with a slow heating (1 °C s^−1^ or less) gives a preferential activation of the C-fibres (thought to be most important for peripheral opioid receptors) and the best evaluation of the second pain. The pain mainly originates from deep tissue group III and IV afferents. However, potentials elicited by mechanical stimulation are mainly the result of a cortical response to Aδ fibres, evaluating the first and fast pain [43].

Second, the results also showed that negative expectations led to poorer perceptual processes, because both groups of negative expectations resulted in reduced detection of both cold and heat. These findings were also observed in the no-expectation control group. Neither positive expectation group showed any improvement, nor did they have reduced detection of thermal thresholds. Perception is a high-level and inferential cognitive process that results from the integration of prior expectations and sensory input [44,45]. Prior expectations can affect how sensory inputs will be processed and can also alter the sensory representation in the sensory cortices [46,47]. Kok et al. [47] found that the representation of a direction in the visual cortex correlated more strongly to the patient’s subjective perceptual report than to the true presented direction. Perception is characterised by dynamic alternative feedback and feedforward processing of information between the sensory cortices and high-level areas, which continuously updates the perceptual process [48]. Lim et al. [49] found that in situations of high levels of prediction errors, or mismatches between prior expectations and sensory inputs, there was increased activation in cognitive-related areas, such as the dorsolateral, orbital, ventrolateral or ventromedial prefrontal cortex, or the inferior parietal lobule, and a decrease in sensory areas, such as the second somatosensory cortex, posterior insula, or middle insula. Negative expectations (expectations of a punishment or poor self-efficacy) might have exerted a top-down influence on thermic perceptual processing.

Finally, we use HR as an indirect measure of neurovegetative activity based on previous works [1,50]. An increase in heart rate indicates activation of the sympathetic excitatory system. MI leads to an activation of the excitatory sympathetic system [13,51], and this activation is based on a preparation phase in which the activation of the sympathetic excitatory system happens to a near effort and, therefore, to a close energy expenditure in physiological processes which will take place in order to face said metabolic changes produced by the voluntary movement itself. In addition, several hypotheses were described regarding the notion that the sympathetic excitatory system not only has the quantitative objective of providing energy to the muscle effectors, but that it also qualitatively and specifically designs and adapts the parameters on demand in an attempt to save the energy provided for each precise motor action [13].

### Study Limitations

The study included an asymptomatic population; thus, their expectations and motivation to find a way to perceive pain relief might be different from a population with musculoskeletal pain. Future studies should replicate these protocols on patients with painful conditions. We did not monitor the participant’s beliefs about the outcome expectations, self-efficacy expectations or intervention efficacy after the intervention and assessments, which could have influenced the expectation modulation effect. In addition, explaining the study procedure before starting is something that may affect the results obtained. This is an intrinsic limitation in expectation studies but should still be reported as an important limitation. The intervention took too few days and was too short in duration. There is no questionnaire or ex-post interview to identify how subjects identified expectations and how they might have influenced the outcome. Finally, there is a need to measure the cognitive state of the subjects in this study; the attentional factor is a determining factor.

## 5. Conclusions

The results obtained showed, first, that positive expectations have a minimal influence on the improvement of heat and mechanical pain detection thresholds. It appears that the HPT is more robust than the PPT, and therefore is less susceptible to modulation by expectations. Positive and non-expectancy groups showed an autonomic activation. Secondly, the results also showed that negative expectations worsened perceptual processes.

## Figures and Tables

**Figure 1 ijerph-19-11878-f001:**
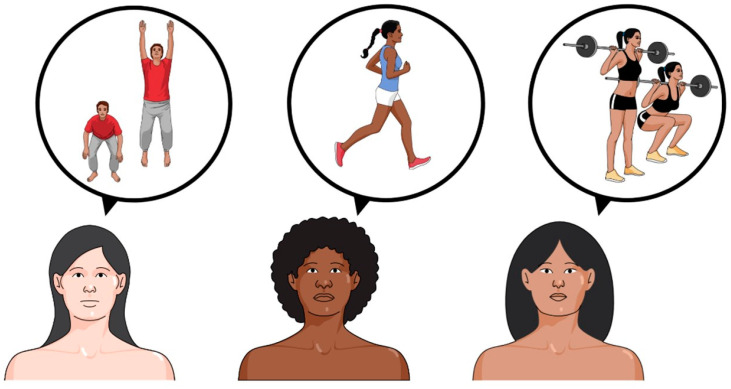
An illustration of the intervention.

**Figure 2 ijerph-19-11878-f002:**
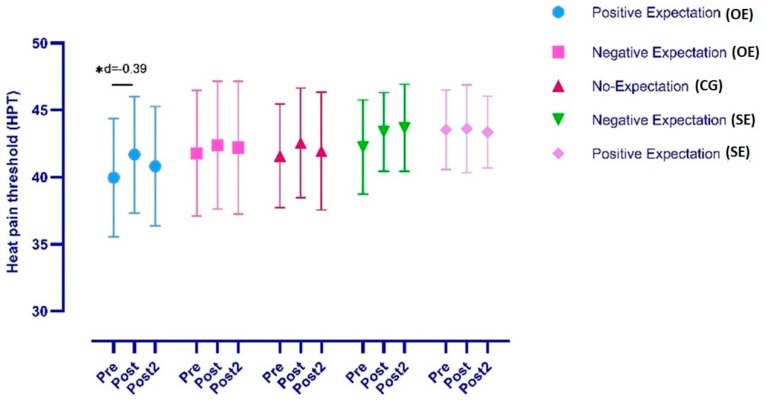
Differences in heat pain threshold between groups and time–group interaction, * *p* < 0.05.

**Figure 3 ijerph-19-11878-f003:**
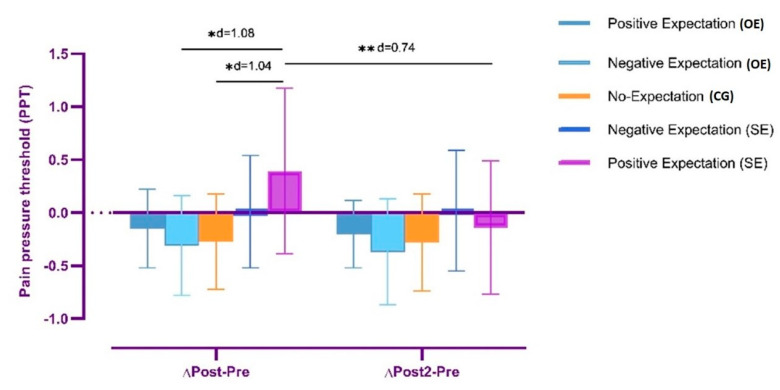
Differences in proximal pain pressure threshold between groups and time–group interaction, * *p* < 0.05, ** *p* < 0.001.

**Figure 4 ijerph-19-11878-f004:**
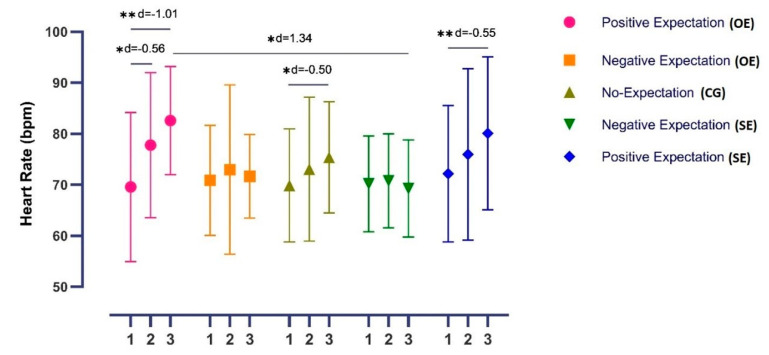
Differences in heart rate between groups and time–group interaction, * *p* < 0.05, ** *p* < 0.001.

**Table 1 ijerph-19-11878-t001:** Descriptive sociodemographic data and baselines measures.

Measures	OE+ (*n* = 15)	OE− (*n* = 15)	CG (*n* = 15)	SE− (*n* = 15)	SE+ (*n* = 15)	*p*-Value
**Age (years)**	29.53 ± 11.46	25.73 ± 8.33	26.13 ± 8.09	21.93 ± 1.87	23.87 ± 7.08	0.124
**BMI (kg/m^2^)**	23.37 ± 2.72	22.06 ± 1.86	23.20 ± 2.91	21.20 ± 1.81	21.98 ± 1.97	0.065
**MIQR-total**	48.40 ± 5.55	45.93 ± 7.23	45.53 ± 8.87	44.33 ± 7.74	44.13 ± 6.03	0.499
**MIQR-K**	23.93 ± 3.19	22.93 ± 4.44	21.73 ± 5.89	21.60 ± 5.04	21.13 ± 4.12	0.460
**MIQR-V**	24.47 ± 3.50	23.00 ± 3.91	23.80 ± 4.64	22.73 ± 4.48	23.00 ± 3.46	0.753
**Synchronisation**	0.95 ± 0.18	0.88 ± 0.12	1.51 ± 1.96	1.15 ± 0.18	1.15 ± 0.32	0.360
**IPAQ-Total**	3656.70 ± 3782.29	2630.23 ± 2300.34	2570.80 ± 1410.93	5415.66 ± 1200.61	2441.1 ± 1356.60	0.590
**Cold detection threshold (°C)**	30.58 ± 1.11	30.34 ± 1.30	30.65 ± 0.84	23.53 ± 1.70	29.85 ± 1.97	0.164
**Warm detection threshold (°C)**	34.33 ± 2.34	34.24 ± 1.12	33.69 ± 0.71	34.51 ± 0.73	34.47 ± 0.88	0.435
**Heat pain threshold (°C)**	39.97 ± 4.42	41.79 ± 4.69	41.62 ± 3.85	42.25 ± 3.51	43.55 ± 2.96	0.183
**Quadriceps femoris pressure pain threshold (kg/cm^2^)**	2.14 ± 1.60	2.18 ± 1.41	2.10 ± 0.98	3.85 ± 1.28	3.28 ± 0.89	<0.001 *
**Upper trapezius pressure pain threshold (kg/cm^2^)**	1.70 ± 1.24	1.75 ± 1.07	1.83 ± 0.91	2.06 ± 0.69	1.95 ± 0.59	0.838
**Gender**						
Male	8 (53.33)	8 (53.33)	4 (26.66)	6 (40.00)	6 (40.00)	0.54
Female	7(46.67)	7 (46.67)	11 (73.34)	9 (60.00)	9 (60.00)	
**Education level**						
Secondary	4 (26.66)	0 (0.00)	1 (6.66)	3 (20.00)	1 (6.66)	0.14
College	11 (73.34)	15 (100.00)	14 (93.34)	12 (80.00)	14 (93.34)	
**Working status**						
Student	7(46.67)	8 (53.33)	5 (33.33)	11 (73.34)	11 (73.34)	0.17
Employee	6 (40.00)	7(46.67)	8 (53.33)	4 (26.66)	4 (26.66)	
Unemployed	2 (13.33)	0 (0.00)	2 (13.33)	0 (0.00)	0 (0.00)	

* Statistical significance at *p* < 0.001. Data are expressed as mean (SD) or *n* (%). Abbreviations: BMI—body mass index; CG—control group; IPAQ-total—International Physical Activity Questionnaire, total score; MIQR-K—The Movement Imagery Questionnaire-Revised, Kinaesthesia score; MIQR-V—Movement Imagery Questionnaire-Revised, Visual score; MIQR-total—Movement Imagery Questionnaire-Revised, total score: OE+: positive outcome expectation; OE−—negative outcome expectation; SD—standard deviation; SE+—positive self-efficacy expectation; SE−—negative self-efficacy expectation.

**Table 2 ijerph-19-11878-t002:** Within-group comparisons of CDT, WDT and proximal PPT during mental representation training.

Measure	Group	Pre	Post	Post 10 min	Mean Difference (95%CI); Effect Size (d) *(a) Pre* vs. *Post* *(b) Pre* vs. *Post 10 min* *(c) Post* vs. *Post 10 min*
**Cold Detection Threshold (°C)**	**OE+**	30.58 ± 1.11	29.94 ± 0.67	30.01 ± 0.93	(a) 0.63 (−0.84 to 1.452); d = 0.69 (b) 0.56 (−0.42 to 1.54); d = 0.55 (c) −0.70 (−0.72 to 0.587); d = −0.08
	**OE−**	30.34 ± 1.30	29.11 ± 2.17	29.18 ± 1.84	(a) 1.23 * (0.41 to 2.051); d = 0.68 (b) 1.15 * (0.17 to 2.14); d = 0.72 (c) −0.07 (−0.733 to 0.581); d = −0.03
	**CG**	30.65 ± 0.84	29.30 ± 2.14	29.29 ± 2.08	(a) 1.34 * (0.53 to 2.16); d = 0.83 (b) 1.35 * (0.37 to 2.34); d = 0.85 (c) 0.00 (−0.64 to 0.66); d = 0.00
	**SE+**	29.84 ± 1.97	30.15 ± 1.49	29.59 ± 1.42	(a) −0.30 (−0.51 to 1.12); d = −0.17 (b) 0.25 (−0.72 to 1.24); d = 0.14 (c) 0.56 (−0.09 to 1.22); d = 0.38
	**SE−**	29.53 ± 1.70	28.48 ± 1.99	28.54 ± 2.12	(a) 1.05 * (0.23 to 1.87); d = 0.56 (b) 0.98 * (0.00 to 1.96); d = 0.51 (c) −0.06 (−0.72 to 0.59); d = −0.02
**Warm Detection Threshold (°C)**	**OE+**	34.33 ± 2.33	34.72 ± 1.43	34.57 ± 1.20	(a) −0.38 (−1.20 to 0.43); d = −0.20 (b) −0.23 (−1.20 to 0.73); d = −0.12 (c) 0.14 (−0.45 to 0.75); d = 0.11
	**OE−**	34.24 ± 1.12	34.87 ± 2.00	35.94 ± 2.18	(a) −0.63 (−1.45 to 0.19); d = −0.38 (b) −1.70 * (−2.66 to −0.73); d = −0.98 (c) −1.06 * (−1.67 to −0.46); d = −0.51
	**CG**	33.69 ± 0.71	34.37 ± 0.70	34.80 ± 1.32	(a) −0.68 (−1.51 to 0.13; d = −0.96 (b) −1.11 * (−2.084 to −0.15); d = −1.04 (c) −0.43 (−1.03 to 0.17); d = −0.40
	**SE+**	34.47 ± 0.88	35.09 ± 0.76	35.20 ± 0.74	(a) −0.61 (−1.43 to 0.20); d = −0.75 (b) −0.72 (−1.69 to 0.23); d = −0.89 (c) −0.11 (−0.72 to 0.49); d = −0.14
	**SE−**	34.51 ± 0.73	35.48 ± 1.65	36.01 ± 2.49	(a) −0.97 * (−1.79 to −0.15); d = −0.76 (b) −1.50 * (−2.47 to −0.54); d = −0.81 (c) −0.53 (−1.14 to 0.075); d = −0.25
**Upper Trapezius Pain Pressure Threshold (Kg/cm^2^)**	**OE+**	1.71 ± 1.25	1.71 ± 1.25	1.76 ± 1.44	(a) −0.003 (−0.315 to 0.309); d = −0.003 (b) 0.057 (−0.408 to 0.294); d = −0.044 (c) −0.053 (−0.256 to 0.149); d = −0.037
	**OE−**	1.75 ± 1.07	1.62 ± 1.00	1.69 ± 1.027	(a) 0.134 (−0.178 to 0.446); d = 0.125 (b) 0.069 (−0.286 to 0.420); d = 0.066 (c) −0.065 (−0.267 to 0.138); d = −0.059
	**CG**	1.84 ± 0.92	1.78 ± 0.858	1.80 ± 0.91	(a) 0.061 (−0.251 to 0.373); d = 0.068 (b) 0.032 (−0.319 to 0.383) d = 0.032 (c) −0.029 (−0.231 to 0.174) d = −0.034
	**SE+**	1.95 ± 0.59	2.05 ± 0.59	1.89 ± 0.56	(a) −0.103 (0.−4.15 to 0.209); d = −0.16 (b) 0.052 (−0.299 to 0.403); d = 0.104 (c) 0.155 (−0.48 to 0.357); d = 0.027
	**SE−**	2.06 ± 0.69	2.11 ± 0.70	2.09 ± 0.69	(a) −0.57 (−0.369 to 0.255); d = −0.072 (b) −0.35 (−0.386 to 0.316); d = −0.043 (c) 0.23 (−0.180 to 0.225); d = 0.028

* The mean difference is statistically significant at *p* < 0.05. Abbreviatures: OE+ = positive outcome expectation group; OE− = negative outcome expectation group; CG = control group; SE+ = Positive Self-Efficacy Expectation Group; SE− = negative self-efficacy expectation group.

## Data Availability

Contact the corresponding authors.

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
