# Peer review of "Effects of Self-Efficacy and Outcome Expectations on Motor Imagery-Induced Thermal and Mechanical Hypoalgesia: A Single-Blind Randomised Controlled Trial"

_ijerph, 2022, doi:10.3390/ijerph191911878_

Round 1

Reviewer 1 Report

Authors addressed whether SE and OE modulate the hypoalgesic effect induced by MI. As results, the positive expectations have a minimal influence on the improvement of heat and mechanical pain detection thresholds and the negative expectations worsened perceptual processes. Overall, this is a well-designed study, but there are some points to consider.

1) The activity of the sympathetic nervous system is important in the perception of pain. The author measured the heart rate considering the relevance of the sympathetic nervous system, and in conclusion, it is assumed that the involvement of the sympathetic nervous system is determined according to the expectancy. If so, it would be nice to have a detailed discussion on whether only the heart rate sufficiently reflects the changes in the sympathetic nervous system and what kind of neural mechanism it is thought to be the case.

2) The author mentioned the importance of pain perception of attention. In this study, the variable for attention was included in positive expectations, but it is necessary to explain how it was evaluated as an actual evaluation index.

3) Authors demostrated that positive outcome expectations did modulate the thermic pain threshold, and positive self-efficacy expectations modulated the mechanic pain threshold. If it is an analgesic mechanism related to cognition in the CNS, it is necessary to explain how the difference between the thresholds of heat pain and pressure pain is distinguished.

Reviewer 2 Report

This is an interesting study, of relevance to the management of pain.

I have some comments and recommendations for revision/further clarification.

Abstract

Well written.

Introduction

 Well written.

Materials and Methods

-          Lines 101-103 (Page 4): Please review this sentence for a better understanding.

-          Lines 127-133 (Page 4): This information could be confusing and should be clarified, since some instructions are not consistent with information showed in Figure 1 (e.g. vertical jump). Please review these sentences to this information be consistent with what´s showed in Figure 1.

-          Lines 244-247 (Page 6):  Perhaps you could consider mentioning the 5 groups and the 3 measures in this sentence for a better understanding.

Results

-          Lines 263-264 (Page 6): Please review this sentence. If there were significant differences, p value should be p<0.05.

-          Lines 321-322 (Page 12): Please review this sentence. If there were no significant differences, p value should be p>0.05.

Discussion

-          Lines 373-375 (Page 13): Please review this sentence for a better understanding

-          Lines 437-438 (Page 14): Please review quotation marks in this sentence and include reference if applicable.

Conclusions

Well written.

Round 2

Reviewer 2 Report

The authors did a great job responding to my suggestions. However, the sentence described on lines 440-444 ("MI is capable ...") should be reviewed for a better understanding.
